



# Can models adequately reflect how long-term nitrogen enrichment alters the forest soil carbon cycle?

Brooke A. Eastman[1], William R. Wieder[2,3], Melannie D. Hartman[4], Edward R. Brzostek[1], and William T. Peterjohn[1]

[1]Department of Biology, West Virginia University, Morgantown, West Virginia USA
[2]Climate and Global Dynamics Laboratory, National Center for Atmospheric Research, Boulder, Colorado, USA
[3]Institute of Arctic and Alpine Research, University of Colorado Boulder, Boulder, Colorado, USA
[4]Natural Resource Ecology Laboratory, Colorado University, Fort Collins, Colorado, USA

*Correspondence to*: Brooke Eastman (brooke.eastman@mail.wvu.edu)

**Abstract.** Changes in the nitrogen (N) status of forest ecosystems can directly and indirectly influence their carbon (C) sequestration potential by altering soil organic matter (SOM) decomposition, soil enzyme activity, and plant-soil interactions. However, model representation of linked C-N cycles and SOM decay are not well-validated against experimental data. Here, we use extensive data from the Fernow Experimental Forest long-term, whole-watershed N fertilization study to compare the response to N perturbations of two soil models that represent decomposition dynamics differently (first-order decay versus microbially-explicit reverse Michaelis-Menten kinetics). These two soil models were coupled to a common vegetation model which provided identical input data. Key responses to N additions measured at the study site included a shift in allocation to favor woody biomass over belowground carbon inputs, reductions in soil respiration, accumulation of particulate organic matter (POM), and an increase in soil C:N ratios. The vegetation model did not capture the often-observed shift in allocation with N additions, which resulted in poor predictions of the soil responses. We modified the plant C allocation scheme to favor wood production over fine root production with N additions, which significantly improved the vegetation and soil respiration responses. To elicit an increase in the soil C stocks and C:N ratios with N additions, as observed, we also modified the decay rates of the particulate organic matter (POM) in the soil models. With all of these modifications, only the microbially explicit model captured a positive soil C stock and C:N response in line with observations. Our results highlight the importance of accurately representing plant-soil interactions, such as rhizosphere priming, and their responses to environmental change.

## 1 Introduction

Northern temperate forests are a globally important carbon (C) sink (Pan et al., 2011; Friedlingstein et al., 2022), but are experiencing rapid changes to their environment that could impact C sequestration rates. Predicting forest responses to environmental change over decadal time scales (or longer) is a challenge that requires the integration of long-term experimental manipulations and models that can detect and simulate changes in ecosystem patterns and processes. For example, many temperate forests have received decades of N deposition from the combustion of fossil fuels, which likely released them from N limitation and contributed to significant C sequestration (Vitousek and Howarth, 1991; Litton et al., 2007; Thomas et al.,



2010; Vicca et al., 2012; Du and de Vries, 2018). Additionally, many N enrichment studies report reductions in soil respiration rates and an accumulation of soil C, which are likely driven by plant reductions in belowground C allocation and lower soil microbial and enzyme activity (Janssens et al., 2010; Schulte-Uebbing and de Vries, 2017; Du and de Vries, 2018). While

most existing models capture the enhancement in plant productivity with N additions, they fail to capture changes in plant C allocation or the reduction in soil respiration fluxes since these fluxes are represented by a positive relationship to plant productivity and litter inputs (Koven et al., 2015; Wieder et al., 2019b; Jian et al., 2021). This shortcoming is especially concerning because, as N deposition declines and forest soil recover, the C that accumulated in these soils may become vulnerable to decomposition and loss. Furthermore, the response of soil heterotrophic respiration to global change will likely

determine the overall magnitude of the land C sink (Bond-Lamberty et al., 2018). Thus, to create meaningful emission reduction targets and mitigate climate change, it is of high priority to predict the drivers and fate of the soil C stock under global change scenarios.

Recent theoretical advancements in the understanding of soil organic matter (SOM) formation and destabilization offer a framework for improving the representation of soil C and N cycling in models (Cotrufo et al., 2015; Lehmann and Kleber,

2015; Sokol et al., 2019). These emerging views highlight how plant productivity and belowground C allocation interact with soil microbial community composition and activity to regulate soil C persistence and heterotrophic respiration fluxes. Nonetheless, the Earth System Models (ESMs) used to predict future C cycles and inform global change policy do not explicitly represent microbial physiology and are limited in their abilities to predict SOM dynamics under environmental change (Wieder et al., 2015b; Varney et al., 2022). Instead, these models typically represent soil C turnover as a linear process with first-order

decay dynamics, and soil C formation is directly related to soil C inputs.

Recently, significant effort has gone towards incorporating explicit microbial communities and microbial physiology into soil models, which may improve the predictive ability of these models—especially under future conditions of environmental change—by incorporating additional mechanisms in the soil C cycle (Sulman et al., 2018; Wieder et al., 2013). For example, by explicitly representing microbial physiology, these models can simulate changes in the temperature sensitivity of

decomposition and soil heterotrophic respiration as the microbial community shifts or microbial growth efficiency acclimates to soil warming (Wieder et al., 2013). Additionally, models are being structured to capture the process of priming that occurs when fresh soil inputs lead to increased microbial demand for nutrients and, thus, accelerated microbial growth and decomposition of SOM (Sulman et al., 2014; Guenet et al., 2013). Furthermore, reductions in microbial activity and shifts in community composition with N additions may contribute to the widely observed reduction in soil respiration with experimental

N additions (Ramirez et al., 2012; Moore et al., 2021; Carreiro et al., 2000), and microbial models may have an advantage over first-order decay models at predicting this response and the downstream impacts this has on soil C storage and cycling. However, few studies have compared the responses of first-order versus microbial models to N perturbations. Therefore, there





is a need to combine modelling and empirical efforts to assess model performance in response to N additions, and to identify any potential benefits of including an explicit representation of microbes and microbial processes (Wieder et al., 2019b).

In this study, we compared how implicit and explicit representations of microbial activity influence ecosystem biogeochemical projections under conditions of elevated N deposition. We evaluated model performance with the results from a 30-year, whole-watershed, N addition field experiment at the Fernow Experimental Forest (Fernow Forest) in West Virginia, USA. The duration and spatial scale of this field experiment provides a unique opportunity to evaluate model assumptions about soil biogeochemical responses to N enrichment. Observations from this long-term field manipulation found that N additions
stimulated aboveground wood production and reduced total belowground C flux (Eastman et al., 2021). Furthermore, this reduced belowground C allocation likely caused a reduction in soil microbial activity as observed through a decrease in soil respiration and leaf litter decomposition, lower rates of ligninolytic enzyme activity and mycorrhizal colonization, and an accumulation of particulate organic matter (POM) in surface mineral soils (Carrara et al. 2018; Eastman et al. 2021, 2022). These responses are observed at other N addition studies, as well, and may be difficult to capture with a first-order, linear
decay soil model because they are driven by shifts in microbial activity and plant-soil interactions—mechanisms not represented in microbially-implicit models.

The main objectives of this study were to compare the default model steady-state C stocks to observations from the Fernow Forest, and to compare observations to the results of three 30-year N addition modelling experiments. These three experiments were: (1) default model responses to N additions; (2) modified models that shift plant C allocation with N additions, in
accordance with field observations; and (3) modified models that both shift plant C allocation and slow the decomposition of POM with N additions. We hypothesized that the default models (experiment 1) would respond to N additions with a reduction in N limitation, increase in plant productivity, and subsequent increase in soil respiration and soil C stocks. We hypothesized that shifting model representation of plant C allocation (experiment 2), and thus reducing plant litter inputs to the soil, will lead to better model-observation agreement by reducing soil respiration rates and shifting the microbial community
composition in MIMICS-CN to favor the oligotrophic (K-type) microbes. Finally, we hypothesized that modifying POM decomposition rates (slower, experiment 3) will help the models reflect observed increases in POM abundance and soil C:N ratios with N additions.

## 2 Methods

### 2.1 Site description

The Fernow Experimental Forest (Fernow Forest) is a broadleaf deciduous forest located in the Central Appalachian Mountains near Parsons, WV (39.03° N, 79.67° W). Elevations at the Fernow Forest range from 530-1,115 m with steep slopes between 20-50% grade. The predominant soils at the Fernow Forest are shallow (<1 m) Calvin channery silt loam (*Typic Dystrochrept*)



underlain with fractured sandstone and shale parent material. Mean monthly temperatures range from about -18 °C in January

to about 25 °C in July, and annual precipitation is about 146 cm with an even distribution across seasons (Kochenderfer 2006).

The Fernow Forest is the site of a long-term, whole-watershed, N-addition experiment. N additions to the experimental

watershed catchment area (Watershed 3; 34 ha) were applied annually by aerial applications of 35.4 kg N ha$^{-1}$ yr$^{-1}$ as

ammonium sulfate from 1989-2019 (30 years). The experimental N addition rate was about double the ambient N deposition

measured in throughfall concentrations at the start of the experiment, and about four times the rate of N deposition by the end

of the experiment (https://nadp.slh.wisc.edu/; www.epa.gov/CASTNET). Aerial application of $(NH_4)_2SO_4$ was distributed in

three applications per year to simulate the seasonal, ambient N deposition rates. An adjacent watershed (Watershed 7) of

similar topography and forest age (24 ha) is used as a reference, receiving only ambient N deposition.

The vegetation at the Fernow Forest is classified as mixed mesophytic forest. The fertilized watershed (Watershed 3) was

harvested using selection harvesting and patch-clearcutting from 1958-1968 before being clear-cut in 1970 and allowed to

regrow naturally for 19 years before fertilization treatment began. The adjacent reference watershed (Watershed 7) was clear-

cut in two sections, the upper half in 1963 and lower half in 1966. Following cutting, both sections of the reference watershed

were kept barren with herbicide treatment until 1969 when the vegetation was allowed to regrow. No legacy effects of the

herbicide treatment were observed ten years into regrowth (Kochenderfer and Wendel, 1983). The Fernow Forest has relatively

diverse vegetation, and tree species are similar in both watersheds, dominated by *Prunus serotina, Acer rubrum, Liriodendron*

*tulipifera,* and *Betula lenta;* although, the fertilized watershed has a greater % basal area of *Prunus serotina* and less

*Liriodendron tulipifera* than the reference watershed.

The observational data from the Fernow Forest used in this study were collected over various time scales and locations in the

fertilized and reference watersheds, with most of these data described and summarized by Eastman et al. (2021). In brief, tree

aboveground net primary productivity (ANPP) measurements were estimated from 25 permanent growth plots per watershed,

in which the aboveground biomass of all trees was estimated six times during the 30-year experiment using measurements of

the diameter at breast height and allometric equations. Also at these plots, autumnal fine litterfall was measured annually from

the start of the experiment (1989) through 2015, and in 20 additional plots per watershed from 2015-2017. Fine root biomass

was measured several times throughout the experiment in various sets of plots using soil cores (ranging in depth from 0-10 cm

to 0-45 cm), and fine root production (0-10 cm) was estimated in 2016-2017 using in-growth cores. Soil organic horizon C

and N stocks were measured in 2012 and 2013, and mineral soil C and N stocks were measured from soil pits (0-45 cm depth)

in 2016. Soil respiration was measured at 80 locations per watershed approximately weekly during the growing season and

monthly during the dormant season for two years (2016-2017) using an infrared gas analyzer. Stream inorganic N export has

been monitored at the Fernow Forest from continuous streamflow measurements and weekly or biweekly streamwater

chemistry samples since 1983 by the US Forest Service. Additionally, we used measurements of the partitioning of SOM into

different soil density fractions in the fertilized and reference Fernow Forest watersheds to compare observed versus modelled



SOM distributions and stoichiometry (Eastman et al., 2022). These mineral soil samples were collected at 20 plots per watershed, in four subplots per plot, to a depth of 15 cm.

## 2.2 Soil biogeochemical model testbed description

The soil biogeochemical model testbed, developed by Wieder et al. (2018, 2019b), provides a framework to compare the performance of two structurally different soil C and N biogeochemical models by coupling them to a common vegetation
model. The soil model testbed was originally developed to facilitate the comparison among three structurally distinct soil C models in their abilities to predict global soil C stocks and their responses to environmental change. Two of these models have been recently modified to include the N cycle and its interactions with the C cycle: one first-order soil C and N model, the Carnegie-Ames-Stanford Approach (CASA-CN; Potter et al., 1993; Randerson et al., 1996; Wang et al., 2010); and one microbially explicit soil C and N model, MIcrobial-MIneral Carbon Stabilization (MIMICS-CN) (Wieder et al., 2014, 2015c;
Kyker-Snowman et al., 2020). While both models were developed and parameterized to run at the global scale, the testbed allows these models to be run at single-point scale, for comparisons against site-level, empirical data.

The soil biogeochemical model testbed provided a computational framework for comparing the response to elevated N inputs of a first-order decay model to a microbially-explicit representation of soil biogeochemical cycles. After calibrating these models to our study site (Sect. 2.3), we ran three 30-year N addition experiments that simulated the long-term N addition study
at The Fernow Forest. The first experiment was performed using the default models calibrated to the study site. In the second experiment, we addressed the assumptions in the common vegetation model about fixed plant allocation. And in the third experiment, we tested the mechanism that N additions can directly inhibit enzyme activity and the decomposition of chemically recalcitrant POM.

### 2.2.1 Overview

The soil biogeochemical model testbed was developed to investigate how model structural assumptions and parameterizations influence global-scale soil biogeochemical projections over the historical record and in future climate change scenarios (Wieder et al., 2018, 2019a). The testbed uses common environmental drivers and a shared vegetation model (CASA-CNP) to reduce uncertainties among soil models that are not directly related to their representation nor the parameterization of soil biogeochemical dynamics. The C and N version of the testbed includes the CASA-CN and the MIMICS-CN soil models. Both
models have three litter pools—metabolic, structural, and coarse woody debris (CWD)—and three SOM pools with various turnover times and stoichiometry (Fig. 1). The three SOM pools in CASA-CN and MIMICS-CN, respectively, include (1) a microbial or SOMa (microbially available) pool with fast turnover; (2) a slow or SOMc (chemically protected); and (3) the passive or SOMp (physicochemically protected) pool (Fig. 1). In this study, we equate the relative abundance of the slow: passive (for CASA-CN) and SOMc: SOMp (in MIMICS-CN) to empirical measurements of POM and mineral-associated organic matter (MAOM) from the Fernow Forest.
organic matter (MAOM) from the Fernow Forest.



Key differences between the models are described in previous work (Wieder et al. 2018; 2019), but here we highlight differences in their representation of soil organic matter turnover and stoichiometry. Litter and SOM turnover in CASA-CN occurs via an implicit representation of microbial activity, with decomposition controlled by linear, first-order dynamics. Soil C turnover times are defined by biome- and pool-specific decay constants that are modified by environmental scalars for soil

temperature and soil moisture availability. The stoichiometry for each of the five organic matter pools in CASA-CN is diagnostic (i.e., values are assigned), and are defined by pool- and biome- specific parameter values (Randerson et al., 1996; Wang et al., 2010; Fig 1a). Conversely, turnover of litter and SOM in MIMICS-CN are determined via temperature sensitive reverse Michaelis-Menten kinetics so that organic matter turnover and heterotrophic respiration fluxes are dependent both on the size of the donor (substrate) and receiver (microbial biomass) pools. MIMICS-CN also represents two functionally distinct

microbial communities that correspond to fast/copiotrophic and slow/oligotrophic growth strategies (or r- and K-type communities, $MIC_r$ and $MIC_K$; Fig 1). These microbial communities have different catabolic potential, anabolic traits, C:N ratios, and substrate affinities (Wieder et al. 2015; Kyker-Snowman et al. 2020). The $MIC_r$ functional group requires more N and has a greater affinity for organic matter with lower C:N ($LIT_m$). In contrast, the $MIC_K$ functional group is relatively more efficient and has a greater affinity for organic matter with higher C:N ratios ($LIT_S$). These functional trait differences lead to

varied stoichiometries of the microbial biomass pools, which are parameterized as C:N ratios of 6 and 10 for $MIC_r$ and $MIC_K$, respectively. The stoichiometries of litter and SOM pools, however, are a prognostic feature of the model, influenced by various factors including litter stoichiometry and the response of the microbial community composition. For this study, it was run at the single point encompassing the study site, Fernow Forest.

### 2.2.2 Model forcing and initialization

The CASA-CNP model consists of coupled vegetation and soil models (Randerson et al., 1996; Wang et al., 2010). In the testbed used for this study, both the CASA-CN soil component and the MIMICS-CN soil model are coupled to the CASA-CNP vegetation model component (although here we only represent coupled C-N biogeochemistry above and belowground). The vegetation component of CASA-CNP requires daily meteorological inputs, including air temperature, precipitation, and GPP. Both soil models (CASA-CN & MIMICS-CN) also need inputs for depth-weighted means of soil temperature and liquid

and frozen soil moisture. The CASA-CNP vegetation model calculates net primary productivity, allocation to leaves, wood and roots, vegetation N demand and uptake, and litterfall fluxes. For this study, input data used to run the model were generated from simulations by the Community Land Model, version 5.0, with satellite phenology (CLM 5.0-SP), forced with GSWP3 climate reanalysis for the period 1900-2014 (Lawrence et al., 2019). In contrast, previous work with the testbed used input data from an older version of CLM (CLM 4.5-SP) forced with Cru-NCEP climate reanalysis data (Wieder et al. 2018; 2019).

In the present study, input data beyond 2014 were generated by extending the CLM 5.0-SP simulation with an anomaly forcing (2015-2019) of atmospheric fields from projections made with the Community Earth System Model version 2 (CESM2, see Danabasoglu et al., 2020; for methods, see also Wieder et al., 2015a, 2019b, who used a similar approach with previous versions of CLM and CESM). Briefly, this anomaly forcing cycles over the last decade of the GSWP3 input and applies an



anomaly based on a 3-member ensemble mean from CESM2 simulations that have been archived for the Coupled Model
Intercomparison Project Phase 6 (CMIP6) experiment. This experiment was run under the "high" emissions pathway, SPP3-
70, climate change scenario to generate data from 2015-2100 (http://www.earthsystemgrid.org). For this study we only present
results through 2019.

From these global simulations we extracted data for the grid cell capturing the Fernow Forest, and the daily CLM 5.0-SP
output were then used as input boundary conditions for all simulations presented here. Because we ran the testbed in single-
point mode, the CASA-CNP vegetation model was assigned one plant functional type (PFT) for our experiment: temperate
deciduous forest. Some of the CASA-CNP vegetation parameters were modified to better represent observations at the Fernow
Forest when appropriate empirical observations were available (Table A1). The CASA-CNP vegetation model simulated NPP
and plant litterfall inputs that become inputs to both soil biogeochemical models (CASA-CN & MIMICS-CN). In the carbon-
only version of the testbed, litterfall fluxes seen by CASA-CN and MIMICS-CN biogeochemical models are identical, but
nitrogen limitation reduces NPP in the CASA-CNP vegetation model (Wang et al. 2010), thus providing a feedback between
soil biogeochemical representations and simulated vegetation pools. In all simulations, soil depth was set to 45 cm to allow
for comparison with observations of soil C and N stocks.

Models were spun-up by cycling over meteorological input data (1900-1919) until C and N pools equilibrated. This took a
spin-up period of 6,000 years for MIMICS-CN and 8,000 years for CASA-CN to ensure that soil stocks reached steady state.
We also ran all simulations through a historic period (1900-1988) using transient GSWP3 climate, N deposition taken from
CLM5 simulation (Lawrence et al., 2019), and atmospheric $CO_2$ data from the same period. Results from historic simulations
were compared with observational data from the Fernow Forest and used to complete the site-specific configuration of the
testbed models.

## 2.3 Site-specific configuration of historic simulations

Based on preliminary results, we modified several parameters in the vegetation and soil model components so that historic
simulations (through 1988) better matched observed ecosystem C and N stocks and fluxes at the Fernow Forest (Sect. 2.1;
based on Eastman *et al.* 2021). All vegetation and soil parameter modifications for site-specific configuration are detailed in
Tables A1, A2, and A3, and these modifications are supported by observational data from the long-term experimental data
(Eastman et al., 2021). Briefly, changes in the CASA-CNP vegetation parameters were made to decrease vegetation C stocks
and increase the baseline N limitation in the model, which was defined by a positive NPP response to N additions (Table A1).

Modifications to CASA soil component parameters reduced the soil C:N ratio and total soil C stocks, again better capturing
observed values (Table A2; Eastman et al., 2021, 2022). In contrast, modifications to the MIMICS-CN soil parameters were
needed to increase total soil C:N ratios and total C stocks, to better reflect observed values and reduce model-to-model
differences (Table A3; Eastman et al., 2021, 2022). After both CASA-CN and MIMICS-CN soil model parameters were



calibrated to the Fernow Forest site for the end of the historic period, the models with these calibrated parameters became the "default" models that were used in experimental simulations (1989-2019) that are the focus of this study.

## 2.4 Experimental design: N enrichment experimental simulations

We performed three soil model testbed experiments that simulated the experimental N additions at the Fernow Forest (1989-2019). Similar to historic simulations, experimental simulations used GSWP3 climate and atmospheric $CO_2$ data that was

extended with an anomaly forcing for years 2015-2019. Each experiment consisted of a control simulation with ambient N deposition rates used in CLM 5.0, and a "+N" simulation that received an additional 3.5 g N $m^{-2}$ $y^{-1}$ distributed evenly across every day of the year (Table 1). This annual rate of additional N deposition matched the annual rate of experimental N additions at the Fernow Forest whole-watershed fertilization experiment (Adams et al., 2006). In the first experiment, "default +N," the N perturbations were the only modifications made to the site-calibrated models (Table 1).

The "default +N" simulation did not capture observed responses to N fertilization that included increases in wood biomass, increases in soil C:N ratio, and a reduction in soil heterotrophic respiration (Fig. 2). In the second experiment, "allocation shift +N," we modified the CASA-CNP vegetation model to address assumptions about plant C allocation. It is well established that more nutrient availability leads to less belowground C flux, and thus increases aboveground NPP (Vicca et al., 2012; Litton et al., 2007; Fernández-Martínez et al., 2017), but this dynamic allocation pattern in response to nutrient enrichment is

one that many models do not capture, including CASA-CNP (Wieder et al., 2019b; Thomas et al., 2015). To improve model representation of observed ecosystem responses at the Fernow, and to test our second hypothesis that reduced soil heterotrophic respiration resulted from shifts in plant allocation away from belowground C inputs, we adjusted carbon allocation of vegetation in CASA-CNP (Table 1). We adjusted the fixed allocation scheme in the CASA-CNP vegetation model to shift 10% of GPP C away from roots and to wood production under conditions of +N. Results from the adjusted allocation scheme

experiment are presented here and referred to as "allocation shift + N" models and simulations hereafter.

In the third experiment, "enzyme inhibition +N," we built on the "allocation shift +N" parameterization to test the additional effect of direct enzyme inhibition: the hypothesis that reduced microbial enzyme activity from elevated soil N led to an accumulation of POM and subsequent increase in the mineral soil C:N ratio. In the MIMICS-CN model, this could be approached multiple ways (see Wieder et al., 2015a), but here we focus on the direct effects that N additions may have by

suppressing ligninolytic enzyme activity, which is supported by observations at the Fernow Forest and other sites (Carreiro et al., 2000; Xia et al., 2017; Carrara et al., 2018; Tan et al., 2020). MIMICS-CN includes a transition of chemically protected SOM ($SOM_c$ which we equate with POM) to microbially available SOM ($SOM_a$). This transition from $SOM_c$ to $SOM_a$ in MIMICS-CN follows reverse Michaelis-Menten kinetics but is not parameterized as a function of soil N availability. To represent potential nitrogen inhibition on POM decomposition, therefore, we increased the half saturation constant for the

oxidation of the chemically protected SOM pool during experimental N additions, essentially reducing rates of decomposition





of this pool (Table 1). In CASA-CN, we adjusted the turnover time of the SLOW pool, increasing it by 30%, a conservative estimate based on observed reductions in ligninolytic enzyme activity in the fertilized watershed (Table 1; Carrara *et al.,* 2018). Results from this experiment are presented here and referred to as "enzyme inhibition + N" models and simulations hereafter.

### 2.5 Model-data comparisons

To compare the sensitivity of observed and modelled responses to N enrichment we calculated response ratios for different C and N pools and fluxes following 30 years of N additions. Response ratios were calculated for key observations and model outputs, using the most recent observed values and the annual mean value from the last 10 years of the experimental simulations. Response ratios were estimated by dividing the ambient (control) observed or modelled value by the +N watershed or modelled value. Thus, a response ratio of 1 indicated that there was no effect of N additions on the pool/flux, whereas a

response ratio greater than or less than one indicated an increase or decrease in that flux/pool with N additions.

### 3 Results

### 3.1 Comparison of baseline calibrated models to observations

Baseline models calibrated to the Fernow Forest had overall good agreement of key carbon and nitrogen pools and fluxes in comparison to observations. Table 2 summarizes baseline calibrated model output from the last ten years of the historic

transient simulations (1979-1988) with comparisons to observations. We compare baseline models to the recent measurements from the reference watershed 7 because this was the watershed and time period with the most complete observational data at the site (see Eastman *et al.* 2021). Because of higher-than observed vegetation nitrogen concentrations (especially in wood) represented in the CASA-CNP vegetation model, models had greater aboveground NPP (ANPP) and plant N uptake fluxes than observations, but slightly lower wood C pools. CASA-CNP vegetation model also simulates much larger fine root C pools

than observed (Table 2).  The discrepancy in fine root C pools is in part due to the depth difference in modeled (45 cm) versus observed (15 cm) values, but CASA-CNP still likely is overestimates this total pool (over 3 times observed; Table 2) that is typically concentrated in the first 20 cm of soil (Jobbágy and Jackson, 2000). As intended, calibrated soil pools, mineral soil C:N ratios, and simulated soil respiration  by baseline models were very similar to observations (Table 2). The CASA-CN soil model attributes more of the total soil C to the organic horizon (litter layers) than MIMICS-CN and observations, and MIMICS-

CN predicted slightly lower soil respiration fluxes (Table 2).

### 3.2 Experiment 1: Default model responses to N additions

Both default versions of the models exhibited a positive response in aboveground plant productivity but were not as sensitive to nitrogen additions as observations, as shown by relatively small increases in aboveground NPP (ANPP; Fig. 2a). Because the default version of CASA-CNP uses fixed plant allocation, changes in leaf, wood and root C pools were all positive,

reflecting increases in NPP that were associated with N fertilization. Overall, the vegetation response to N addition was



stronger with the MIMICS-CN soil model than with the CASA-CN soil model. Belowground, both soil models predicted little to no change in soil C stocks, a slight increase in heterotrophic soil respiration, and very slight decreases in soil C:N ratios (Fig. 2b). These modeled soil responses were opposite to what was observed (Fig. 2b) and were due to the overall positive response of NPP and, thus, plant matter inputs to the soil. Therefore, we targeted the plant allocation scheme for the vegetation

allocation shift +N experiment.

### 3.3 Experiment 2: Plant allocation shifts with N additions

To elicit a vegetation response in CASA-CNP that reflected the observed shift in plant C allocation, we modified allocation parameters for fertilized experiments in the CASA-CNP vegetation model. As intended, this "plant allocation shift" modification to CASA-CNP vegetation model improved model-observation agreement through a more positive ANPP

response, enhanced woody biomass C stocks, and reduced fine root production with N additions in both coupled vegetation-soil models (Fig. 2a). This change in the vegetation response influenced soil biogeochemical responses by both models, as well.

Notably, the significant (~20%) reduction in root C inputs to the soil lead to a small reduction in soil respiration (~6%) with N additions in both soil models (Fig. 2b). However, the combination of large reductions in soil C inputs and small reductions

in soil C outputs (respiration), resulted in an overall 3–4% reduction – rather than the observed stimulation – in the total soil C pool with both models (Fig 2b). Soil models diverged in soil stoichiometric response to N additions, with a slight decrease in soil C:N simulated in the CASA-CN "plant allocation shift" experiment — similar to the default experiment —but a very slight increase in soil C:N resulting from the MIMICS-CN "plant allocation shift" experiment, which was more similar to the mean observed response (Fig. 2b). These increases in soil C:N ratio resulting from the plant allocation shift and N additions

in MIMICS-CN coincided with a subtle accumulation of POM (SOMc; Fig. 3).

### 3.4 Experiment 3: Enzyme inhibition of decomposition with N additions

Based on observed increases in light particulate organic matter (POM) and soil C:N ratios with N additions in the surface soil at the Fernow Forest (Eastman et al., 2022), we examined whether the distinct soil models could capture this pattern with an additional modification that reflected a reduction in soil enzyme activity with the elevated N perturbation (Fig. 1; Table 1).

These "enzyme inhibition +N" experiments generated similar plant productivity responses as in the "plant allocation shift +N" simulations (Fig. 2a). By increasing the turnover time of the CASA-CN SLOW pools and reducing the oxidation rate of $SOM_c$ in MIMICS-CN, both models simulated increases in the total soil C stocks that are more consistent with observations (5% and 8%, respectively) and, particularly, in the POM pools (SLOW and $SOM_c$; Figs. 2b, 3). Only in MIMICS-CN, however, did this lead to a more positive response, or increase in, the soil C:N ratio that closely approximated the observed mean value (Fig.

2b). Similar to observations, the positive response of the bulk soil C:N ratio that occurred with N additions was concurrent to an increase in the relative abundance of the POM pools in "enzyme inhibition +N" simulations (Fig. 3). However, the





relationship between the fraction of soil C in POM and bulk soil C:N ratios captured by the models were weak compared to the actual relationship found in samples collected from both watersheds at the Fernow Forest (Fig. 3). The weak relationship between POM abundance and bulk soil C:N ratios was due to the low C:N ratios of the POM pools in CASA-CN and MIMICS-

CN models.

## 4 Discussion

Using a soil model testbed to evaluate model responses to N additions, we found that modifying plant C allocation and soil POM decomposition rates under conditions of elevated N deposition most improved model-observation agreement (Fig. 2). Coupled to a vegetation model with a static allocation scheme, both CASA-CN and MIMICS-CN models captured general

observations of key ecosystem pools and fluxes in the reference watershed (Table 1). However, without modification to the models, they failed to capture some key observed responses to N additions: increased woody biomass production, reduced belowground C allocation, reduced soil respiration, and POM accumulation in surface mineral soil (Fig. 2, default models; Fig. 3). With our model experiments, we show that modifications to plant C allocation that increased the overall turnover time of vegetation C created the greatest improvement of model-observation agreement to N additions (Fig. 2). However, this

modification still failed to adequately capture two important observed responses: an increase the total soil C:N ratio, and an increase in the pool of total soil C.

Further refinement of the models to simulate a direct inhibition of microbial activity moved the soil model predictions closer to mean observed increases in soil C stocks and improved the predicted decreases in soil respiration with N additions (Eastman et al., 2021; Fig. 2b). Furthermore, this modification to one of the models (MIMICS-CN) resulted in elevated soil C:N ratios

that matched observed values. Given the widespread occurrence of reduced soil respiration and microbial activity with N additions (Janssens et al., 2010), as well as the importance of this C flux for the future of the land C sink (Bond-Lamberty et al., 2018), validating model assumptions against long-term experimental data is a necessary step to improve our predictions of the land C sink to global change.

### 4.1 Implications of a fixed allocation vegetation model

Our model efforts suggest that capturing the shifts in plant C allocation in response to N additions is the most impactful way to improve the modeled N fertilization response. The default parameterization with static plant C allocation were not sensitive to N additions, suggesting that the models underestimate N limitation by plants (Fig. 2). When we modified the fixed allocation scheme in the CASA-CNP vegetation model under elevated N inputs, both models captured the often observed increase in wood production and reduction in belowground carbon flux (root inputs) with N additions (de Vries et al., 2014; Fernández-

Martínez et al., 2014; Frey et al., 2014; Zak et al., 2008). We note this model experiment was a post-hoc modification to the CASA-CNP allocation parameters for fertilized simulations, but it underscores the importance of future work to develop more





robust parameterizations that moderate plant C allocation as a function of ecosystem fertility status (e.g., Parton et al. 2010, Shi et al. 2016). Such developments in models that do not already account for dynamic allocation shifts are critical for making more accurate projections of plant NPP responses to global change drivers, and the role of terrestrial ecosystem in sequestering

atmospheric $CO_2$ (Shi et al., 2019). Notably, with improved vegetation responses to fertilization in our "plant allocation shift +N" simulations, reduced soil heterotrophic respiration rates followed from reductions to belowground C inputs in both the model and experimental results (Fig. 2b). The ability of the "plant allocation shift +N" model experiment to capture the observed responses at the Fernow mirrors other recent model-experiment integration efforts. For example, a recent model-data synthesis of forest responses to elevated $CO_2$ showed that the models that performed best had dynamic representations of C

allocation that were responsive to water and nutrient availability (De Kauwe et al., 2014). As such, there remains a clear need to prioritize models that employ dynamic allocation approaches and parameterizations based on data syntheses.

One shortcoming of our model efforts was the inability to represent a meaningful rhizosphere priming response in the control simulation that would lead to reduced priming in the elevated N simulations. High soil N availability encourages shifts in plant nutrient acquisition strategies by reducing belowground C flux to mycorrhizae that is typically required for nutrient acquisition

(Gill and Finzi, 2016; Eastman et al., 2021). At our study site, shifts in nutrient acquisition strategy and C allocation led to reduced mycorrhizal colonization, reduced rates of SOM decomposition, and an accumulation of POM. Future modelling attempts could implement a root exudate flux in a way that reflects C allocated to the rhizosphere and mycorrhizal community (e.g., K-type microbes, or a new pools of microbes) and also incorporates a N component of the exudate flux to stimulate microbial growth and activity. Some ecosystem models do consider plant exudate inputs to the soil that prime the rhizosphere

community for N acquisition (e.g., FUN-CORPSE; Sulman et al. 2017). Because N acquisition comes with a C cost, such a transactional representation of N acquisition and uptake may better predict the plant C allocation response to elevated N inputs (Thomas et al., 2015). As we were not able to address this shortcoming in this study, we targeted decomposition rates and simulated the direct inhibition to decomposition by N additions instead.

### 4.2 Enzyme inhibition and soil C accumulation

Despite significant improvements brought forth by the modified plant allocation scheme, the strong reductions in soil C inputs (by fine roots) with N additions outweighed the more subtle reductions in soil C outputs (heterotrophic respiration) and led to an overall reduction in soil C stocks, contrary to observations (Fig. 2). Subsequently, we tested the enzyme inhibition hypothesis, reducing decomposition rates of POM pools with N addition. Augmented N likely increases the turnover time of the POM pool through reduced oxidative enzyme activity and less microbial priming (Von Lützow et al., 2008; Eastman et al.,

2022; Craine et al., 2007; Chen et al., 2018). While our modeling efforts did successfully increase the soil C stocks in both CASA-CN and MIMICS-CN, it only led to an increase in soil C:N ratio response in the MIMICS-CN model (Fig. 2b). The ~8% increase in soil C with N addition predicted by the MIMICS-CN "enzyme inhibition +N" model was similar to the mean enhancement in surface mineral soil (0-15 cm) at the Fernow Forest (~11%; Eastman et al., 2021), as well as increases in



surface soil C stocks at other long-term N addition experiments (Frey et al., 2014; Zak et al., 2008). The CASA-CN model
predicted a more moderate enhancement in soil C stocks (5%) with N additions. The first order decay dynamics in models like
CASA-CN more closely links soil C input fluxes to soil C stocks (see Friend et al., 2014; Koven et al., 2015), so a stronger
decrease in fine root production in CASA-CN likely resulted in a weaker soil C stock response to N additions.

Beyond accurately predicting changes in the *total* soil C stocks and fluxes, the *distribution* of SOM among POM and MAOM
pools is of high importance to the future land C sink (Lavallee et al., 2020; Whalen et al., 2022). Changes in the distribution
of these SOM pools may impact overall soil stoichiometry (Mikutta et al., 2019, Eastman et al., 2022), which drives important
soil C and nutrient cycling processes, such as net N mineralization rates (Aber et al., 2003; Venterea et al., 2004). An increase
in the relative proportion of POM constituting SOM stocks in the fertilized watershed at the Fernow Forest raises compelling
questions about the future of C and N accumulations due to chronic N additions in a changing world. For example, how will
N-induced increases in the relative importance of POM impact forest recovery from N deposition and progressive N limitation
under elevated $CO_2$ conditions (Craine et al., 2018; Groffman et al., 2018; Norby et al., 2010)? Indeed, a recent global analysis
by Hartley et al. (2021) found evidence for greater vulnerability of POM decomposition under conditions of soil warming
compared to MAOM.

MIMICS-CN offers an advantage over CASA-CN because of the diagnostic soil stoichiometry and more mechanistic
decomposition dynamics, which allowed for greater shifts in soil organic matter composition (i.e., POM accumulation) and
C:N ratios compared to CASA-CN. Capturing shifts in bulk soil C:N requires representation of multiple pathways of SOM
formation, which MIMICS-CN includes, such as microbial biomass turnover *and* the direct physical transfer of litter-derived
organic matter that has bypassed microbial decomposition (Cotrufo et al. 2019; Cotrufo et al. 2015). Observed C:N ratios of
POM at the Fernow study site were ~25, but the C:N ratios in CASA-CN and MIMICS-CN were between ~14-20. In CASA-
CN, the prescribed C:N ratio of POM is lower than observed, leading to very small changes in total bulk soil C:N even as the
fraction of SOM in the POM pool increases. Thus, even if the POM turnover time in CASA-CN was further increased, it would
still not capture the mean observed increase in soil C:N with POM accumulation (Fig. 3). In MIMICS-CN, the increase in total
soil C:N was mainly driven by the relative increase in the fraction of POM and decrease in low C:N MAOM fraction (not
shown). Still, the increase in bulk soil C:N with POM accumulation in MIMICS-CN was not as strong as observations suggest
(Fig. 3). This was likely due to the mechanism targeted in the "enzyme inhibition +N" experiment: reducing the oxidation of
SOMc to SOMa. In MIMICS-CN, most litter inputs pass through a microbial pool, and SOM pools are mostly made up of
microbial necromass—with a lower C:N ratio—though a small amount of litter inputs bypass microbial pools (Fig. 1). This
underscores challenges in assessing plant vs. microbial contributions to SOM formation and persistence (Whalen et al., 2022).
Under conditions of elevated N inputs, it is thought that more litter inputs bypass microbial decomposition. Therefore, the
direct transfer of litter inputs to soil pools is a key pathway that may better achieve observed responses of the SOM stocks and
composition to N amendments in MIMICS-CN.



While the microbial explicit foundation of MIMICS-CN holds promise, there still appears to be uncertainties in how plant-soil interactions and their responses to environmental change should be presented and parameterized in the models. For example, our post-hoc adjustment of decay rates with N enrichment emphasize the need to develop model parameterizations that capture phenomena in real-world ecosystems. Given the widespread empirical evidence for a reduction in lignin-degrading enzyme

activity with elevated N inputs (Treseder, 2004; Pregitzer et al., 2008; Frey et al., 2014; Carrara et al., 2018), and the resulting impacts on soil stoichiometry (Chen et al., 2018), additional efforts to improve mechanistic representations of decomposition parameterizations with available data should be a focus area for future model improvement. Such responses, however, are nuanced across ecosystems as Rocci *et al.* (2022) found no consistent changes in soil stoichiometry with nutrient additions in grassland ecosystems, despite an increase in the relative fraction of POM compared to MAOM. Additionally, the microbial

community composition, as approximated by the relative abundance of $MIC_r:MIC_K$ simulate by MIMICS-CN, was not sensitive to N additions or shifts in plant allocation and inputs (not shown). By contrast, N addition experiments in forests ecosystems have found reductions in fungal decomposer biomass, reduced ligninolytic enzyme activity, (Frey et al., 2014; Argiroff et al., 2019), and a shift in community function with reduced ability to decompose recalcitrant SOM (Ramirez et al., 2012)—including at our study site (Carrara et al., 2018; Moore et al., 2021). In these studies, this shift in microbial community

and function results in accumulation of SOM. Models like MIMICS-CN need further development to accurately represent these changes in community composition as resource availability and stoichiometry shift to simulate the downstream effects on soil biogeochemistry. Currently, microbial communities in the model may have too great of access to SOM and litter inputs, resulting in more rapid decomposition rates and lower C:N ratios of SOM pools than is often observed (see also Kyker-Snowman et al. 2020). Specifically, constraining carbon use efficiencies (CUE), nitrogen use efficiencies (NUE), and C:N

ratios for microbial communities against data and observations is warranted to capture their responses to environmental changes.

**Conclusions**

The two models tested in this study showed that targeted modifications, informed by results from a long-term experiment, significantly improved their ability to capture some key ecosystem responses to N additions: notably, a shift in plant C

allocation to favor wood biomass over belowground allocation, decreased soil respiration, and an accumulation of POM with high C:N ratios (Eastman et al., 2021, 2022). Our results also suggest that a more microbially explicit model has a greater potential to capture these complex responses to N enrichment and predict ecosystem recovery from N additions compared to first-order relationships that have been commonly used in the past. However, there are still key mechanisms driving the response of the forest soil C cycle to N additions – such as direct enzyme inhibition, reduced rhizosphere priming, and shifts

in microbial community comopsition – that should be better represented in future modelling efforts.



**Appendices**

**Table A1.** Parameter modifications made to CASA-CNP vegetation model for site-specific configuration during spin-up and historical runs.

| CASA-CNP Vegetation Model | | | | |
|---|---|---|---|---|
| **Parameter** | **Default** | **Modified** | **Source** | **Description** |
| Fine root mean age *(years)* | 10 | 1.45 | Eastman & Peterjohn, *upublished data* | reduce fine root biomass to better match observations |
| Allocation of GPP C (leaf, wood, froot) | 0.3, 0.2, 0.5 | 0.3, 0.3, 0.4 | Eastman *et al.*, 2021 | Increase wood C stocks and decrease fine root C stocks |
| Wood respiration *(year$^{-1}$)* | 6 | 3 | Eastman *et al.*, 2021 | Adjust NPP and wood C stocks to match observed |
| Leaf C:N | 50 | 42 | Eastman *et al.*, 2021 | Match observed |
| Leaf N:C (min, max) | 0.02, 0.024 | 0.0222, 0.02439 | | Capture modified target leaf C:N |
| Fine root C:N | 41 | 35 | Adams, 1991 | Match observed |
| Fine root N:C (min, max) | 0.02439, 0.029268 | 0.025, 0.032258 | | Capture modified target fine root C:N |
| N:C ratio CWD (max) | 0.006857 | 0.00625 | Eastman *et al.*, 2021 | Increase C:N of CWD, decrease N availability |
| N leach rate *(g N m$^{-2}$ y$^{-1}$)* | 0.01 | 0.15 | Adams *et al.*, 2006 | Closer to observed rates; Increase N limitation under ambient N deposition |
| Max fine litter pool *(g C m$^{-2}$)* | 887 | 1527 | Greatest value of all CASA PFTs | Increases N limitation |





| Max CWD pool (g C m⁻²) | 1164 | 1918 | Greatest value of all CASA PFTs | Increases N limitation |
|---|---|---|---|---|
| xkNlimiting (min, max) | 0.5, 2 | 3.4, 5.6 (CASA only) | | Increases N limitation in CASA model, to be more similarly N limited as the MIMICS-CN model, by increasing the amount of soil N needed to maintain plant N uptake rates. |


**Table A2.** Soil parameter modifications made to CASA-CN for site-specific configuration during spin-up and historical runs. Soil C and N stocks and C:N ratios were compared against observations from Eastman *et al.* (2021, 2022).

| CASA-CN | | | |
|---|---|---|---|
| **Parameter** | **Default** | **Modified** | **Justification** |
| MIC soil pool mean age *(years)* | 0.137 | 0.30688 | Decrease total soil C:N ratio |
| SLOW soil pool mean age *(years)* | 5 | 3 | Decrease SLOW soil pool, total soil C:N ratio, and soil C and N stocks |
| PASSIVE soil pool mean age *(years)* | 222.22 | 621 | Increase PASSIVE soil pool; decrease total soil C:N ratio |
| MIC pool C:N (target, min, max) | 8, 6.69, 8 | 7, 6, 10 | Decrease total soil C:N ratio |
| SLOW pool C:N (target, min, max) | 30, 16.2, 30 | 14, 12, 16 | Decrease total soil C:N ratio |
| PASSIVE pool C:N (target, min, max) | 30, 16.2, 30 | 13, 10, 15 | Decrease total soil C:N ratio |





445**Table A3.** Soil parameter modifications made to MIMICS-CN for site-specific configuration during spin-up and historical runs. Default
values are those used by Kyker-Snowman *et al.* (2020). Some parameters used were sourced from the C-only global simulation of the
tesbed (Wieder *et al.,* 2015), and denoted as such. Soil C and N stocks and C:N ratios were compared against observations from
Eastman *et al.* (2021, 2022).

| MIMICS-CN | | | | |
|---|---|---|---|---|
| **Parameter** | **Default** | **Modified** | **Description** | **Justification** |
| $a_V$ | $4.8 \times 10^{-7}$ | $8 \times 10^{-8}$ | Tuning coefficient | Increases decomposition rates of all pools; Wieder *et al.,* 2015 |
| $K_{slope}$ <br> $ln(mg\ C\ cm^{-3})\ °C^{-1}$ | 0.017-0.027 | 0.025 | Regression coefficient | Wieder *et al.,* 2015 |
| $a_K$ | 0.5 | 10 | Tuning coefficient | Wieder *et al.,* 2015 |
| $V_{mod}$ (k2) | 2.25 | 2.5 | Modifies $V_{max}$ for fluxes from LITs to MICk | Increases decomposition of structural litter |
| τ_r <br> $(h^{-1})$ | 0.00024, <br> 0.3 | 0.000624, <br> 0.6 | Controls r-type microbial biomass turnover rate | Increases turnover of r-type microbial biomass |
| τ_k <br> $(h^{-1})$ | 0.00011, <br> 0.1 | 0.000288, <br> 0.1 | Controls k-type microbial biomass turnover rate | Increases turnover of K-type microbial biomass |
| τ Mod <br> (min, max) | 0.6, 1.3 | 1, 1 | Modifies microbial biomass turnover rate | Wieder *et al.,* 2015; (no modification) |
| $f_P$ (r) | 0.015, 1.3 | 0.2, 1.3 | Fraction of τ (r) partitioned to SOMp $0.2 \times e^{1.3(fclay)}$ | Increases fraction of r-type microbial biomass partitioned to SOMp |
| $f_P$ (k) | 0.01, 0.8 | 0.2, 0.8 | Fraction of τ (k) partitioned to SOMp $0.2 \times e^{0.8(fclay)}$ | Increases fraction of K-type microbial biomass partitioned to SOMp (Wieder *et al.,* 2015) |





| $D$ ($h^{-1}$) | $1.0 \times 10^{-6}$, -4.5 | $1.0 \times 10^{-6}$, -1.5 | Desorption rate from SOMp to SOMa $10^{-6} \times e^{-1.5(f\text{clay})}$ | Increase desorption rate from SOMp to SOMa (Wieder et al., 2015) |
|---|---|---|---|---|
| $f_I$ (met) | 0.05 | 0.3 | Fraction of metabolic litter inputs transferred to SOMp | Increase total soil C stocks, increase SOMp |
| $f_I$ (struc) | 0.3 | 0.35 | Fraction of structural litter inputs transferred to SOMc | Increase SOMc, increase total soil C:N ratio |
| $f_{met}$ | 0.85—0.013 | 0.65—0.013 | Partitioning of plant litter inputs to metabolic pool | Reduce fraction of inputs partitioned to metabolic pool (Wieder et al., 2015) |
| NUE (1, 2, 3, 4) ($mg\ mg^{-1}$) | 0.85, 0.85, 0.85, 0.85 | 0.8, 0.7, 0.8, 0.7 | Proportion of mineralized N captured by microbes (1) LITmN or SOMaN to MICrN; (2) LITsN to MICrN; (3) LITmN or SOMaN to MICkN; (4) LITsN to MICkN | By reducing NUE, we reduced the microbial competitive advantage over plants for N and N limitation. Reducing NUE more for structural litter fluxes increased soil C:N |
| CN_r, CN_k | 6 10 | 8 12 | C:N ratio of r-type microbes C:N ratio of k-type microbes | Increase soil C:N; reduce microbial N demand & N limitation |
| fracDINavailMIC | 0.5 | 0.2 | Fraction of dissolved inorganic N available to microbes | Reduce N limitation by decreasing microbial N uptake |
| Soil Depth ($cm$) | 100 | 45 | Total soil depth | Observed values are measured to a depth of 45 cm |


**Biogeosciences** Open Access

**Discussions**

EGU

**Code and data availability**

Model code is available at https://doi.org/10.5281/zenodo.7636494. Model code, results, analysis code can also be accessed at https://github.com/wwieder/biogeochem_testbed. Model output was analyzed and figures were produced in R (R Core Team, 2020), using packages tidyverse (Wickham et al., 2019), data.table (Dowle and Srinivasan, 2020), stringr (Wickham, 455 2019), and scales (Wickham and Seidel, 2020).

**Author contribution**

All authors contributed to the conceptualization of the project and experimental design. WRW and MDH developed the model code and BE performed the model simulations and analyzed model output with key input from WRW. BAE prepared the manuscript with contributions from all co-authors.

**Competing interests**

The authors declare that they have no conflict of interest.

**Acknowledgements**

This research was funded by a National Science Foundation Long-Term Research and Environmental Biology awards [Grant nos. DEB-0417678, DEB-1019522, and DEB-1455785]. B.E. also received support from National Center for Atmospheric 465 Research Center Advanced Study Program's Graduate Student Fellowship, NCAR is a major facility sponsored by the NSF under Cooperative Agreement No. 1852977. W.R.W. acknowledges support from the NSF (Awards 1926413 and 2031238) and USDA-NIFA (2020-67019-31395).

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





**Tables and Figures**

**Figure Legends**

**Figure 1.** Conceptual diagram of the soil biogeochemical model testbed which includes the (a) CASA-CN vegetation model, (b) CASA-CN soil model, and (c) MIMICS-CN soil model. All pools (boxes) and fluxes (arrows) represent both C and N processes, except for the Inorganic N pools and the fluxes into and out of these pools. Highlighted in yellow are processes that
were modified to test two key hypotheses: (1) modifying C allocation to plant tissues to increase wood production with N additions; and (2) decreasing the turnover time (CASA) or increasing the half-saturation constant (MIMICS) of the slow (CASA) or chemically protected (MIMICS) soil pools under conditions of elevated N (See Table 1).

**Figure 2.** Observed and modeled response ratios of select vegetation (left) and soil (right) pools and fluxes to the three nitrogen
addition experiments. Observations (black circles) show the mean (+/- se) values across 10 plots per watershed from the watershed fertilization study at the Fernow Forest (Eastman *et al.,* 2021). Modelled responses include the annual mean from the last ten years of experimental N additions for CASA-CN (brown) and MIMICS-CN (blue) default models (triangles), modified vegetation allocation models (allocation shift; square), and the modified vegetation allocation and soil decay models (enzyme inhibition; asterisk). The vertical dashed line represents no effect of N additions. Aboveground net primary
productivity (ANPP) is the sum of leaf C flux to soil and annual wood C increment (modeled estimates, only, include coarse roots). Total soil pools include organic and mineral horizons, to a depth of 45 cm for both modelled and observed values. Root flux/TBCF is the total belowground carbon flux of inputs to the soil (modeled values are root turnover inputs and observed values are a mass balance of soil respiration minus leaf C flux). †Observed soil respiration includes autotrophic + heterotrophic, whereas modelled soil respiration includes only heterotrophic.


**Figure 3.** Relationship between the relative proportion of light particulate organic matter (POM; named SLOW and SOMc pools in CASA and MIMICS, respectively) and the C:N ratio of bulk mineral soil in observed (black circles) and modeled (brown=CASA, blue=MIMICS) ambient and +N conditions. Figure adapted from Eastman *et al.* (2022). Observed points represent the mean of four soil samples from the top 10 cm of mineral soil per plot (from 10 plots per watershed). Modeled
estimates are from the entire mineral soil profile (0-45 cm). Linear regression (standard error in gray shading) for observed (solid black) and modelled (dashed) values.





 **Tables**

**Table 1.** Description of model experiments, including model parameter modifications made to test plant C allocation shift and enzyme inhibition hypotheses. Bold text indicates parameter modification.

| Experiment | Model | Control or +N | Plant C allocation (leaf:root:wood) | SLOW pool decay dynamics |
|---|---|---|---|---|
| *Experiment 1* *"Default +N"* | CASA-CN | Control | 0.3 : 0.4 : 0.3 | turnover = 4 y |
| | CASA-CN | +N | 0.3 : 0.4 : 0.3 | turnover = 4 y |
| | MIMICS-CN | Control | 0.3 : 0.4 : 0.3 | KO† = 6 |
| | MIMICS-CN | +N | 0.3 : 0.4 : 0.3 | KO† = 6 |
| *Experiment 2* *"Allocation shift +N"* | CASA-CN | Control | 0.3 : 0.4 : 0.3 | turnover = 4 y |
| | CASA-CN | +N | **0.3 :0.3 : 0.4** | turnover = 4 y |
| | MIMICS-CN | Control | 0.3 : 0.4 : 0.3 | KO† = 6 |
| | MIMICS-CN | +N | **0.3 :0.3 : 0.4** | KO† = 6 |
| *Experiment 3* *"Enzyme inhibition +N"* | CASA-CN | Control | 0.3 : 0.4 : 0.3 | turnover = 4 y |
| | CASA-CN | +N | **0.3 :0.3 : 0.4** | **turnover = 5.33 y** |
| | MIMICS-CN | Control | 0.3 : 0.4 : 0.3 | KO† = 6 |
| | MIMICS-CN | +N | **0.3 :0.3 : 0.4** | **KO† = 9** |

†KO is a scalar to modify Km in the Michaelis-Menten equation for the oxidation of SOMc to SOMa, where MIC is microbial biomass C; Vmax is maximum velocity; SOMc is the SOMc C pool; Km is the half saturation constant.

$$SOMc \rightarrow SOMa = \frac{MIC \; x \; Vmax \; x \; SOMc}{\mathbf{KO} \; x \; Km + SOMc}$$





**Table 2.** Baseline model simulations and observations of the ecosystem pools and fluxes. Model mean (se) values are from the last ten year of the historic transient simulations (1900-1988). Observed mean (se) values are recent measurements (2009-2018) from the reference watershed 7, for which the most complete data were available.

| Ecosystem Pool/Flux | MIMICS-CN model | CASA-CN model | Observation |
|---|---|---|---|
| *Vegetation* | | | |
| GPP (g C m$^{-2}$ y$^{-1}$) | 1342 (68) | 1342 (68) | -- |
| ANPP (g C m$^{-2}$ y$^{-1}$) | 772 (8.2) | 805 (10) | 565 (25) |
| Woody biomass C (g C m$^{-2}$) | 9,300 (5) | 9,605 (8) | 11,475 (634) |
| Fine root C (g C m$^{-2}$) | 497 (1.6) | 521 (2.5) | 152 (31) |
| Plant N uptake (g N m$^{-2}$ y$^{-1}$) | 13.5 (0.2) | 13.8 (0.2) | 7.6 (2.4) |
| *Soil* | | | |
| Leaf litter inputs (g C m$^{-2}$ y$^{-1}$) | 198 (0.8) | 204 (0.7) | 162 (2) |
| Mineral soil C (g C m$^{-2}$) | 8,220 (6) | 6,641 (4) | 8,299 (566) |
| Organic horizon C (g C m$^{-2}$) | 505 (2.2) | 1322 (1.6) | 539 (24) |
| Total soil C pool (g C m$^{-2}$) | 8,725 (11) | 7,963 (0.3) | 8,838 (513) |
| Mineral soil C:N ratio | 13.6 (0.3) | 13.4 (0.3) | 14.4 (1.6) |
| Soil respiration (g C m$^{-2}$ y$^{-1}$) | 754 (7) | 906 (4) | 982 (63) |
| N leaching (g N m$^{-2}$ y$^{-1}$) | 2.1 (0.06) | 1.5 (0.09) | 1.1 (0.06) |






**Figures**




Figure 1. Conceptual diagram of the soil biogeochemical model testbed







**Figure 2.** Observed and modelled response ratios of select vegetation (left) and soil (right) pools and fluxes to the three nitrogen addition experiments.






**Figure 3. Relationship between the relative proportion of light particulate organic matter and the C:N ratio of bulk mineral soil in observed (black circles) and modeled (brown=CASA, blue=MIMICS) ambient and +N conditions.**
