# Peer review of "Can models adequately reflect how long-term nitrogen enrichment alters the forest soil carbon cycle?"

_Biogeosciences, 2023_

## Author Response (AR1)

Dear Dr. Eastman

The three reviewers are generally supportive of the scope of your manuscript, confirm that it matches our special issue, and acknowledge the value of the presented research. Your responses to the reviewers' comments and suggested edits are mostly convincing. However, especially reviewer 3 raises some important and fundamental points that should be addressed more profoundly than you suggest. In particular:

- The "post-hoc" nature of your model modification (allocation fractions, turnover time mimicking enzyme inhibition) should be taken up throughout the manuscript and the applicability and generalisability of the (modified) model explained. For example, I recommend not to call this a "modified the plant C allocation scheme" (Abstract), but a case where where essential parameters of a single model were modified in some simulations (but not in others) to match observations of the perturbed state in an N-fertilisation experiment.

*BAE et al: Thank you for reiterating this point from reviewer 3. Text in the manuscript is edited—in the abstract and throughout—to clearly indicate that "modified models" were rather modified parameterizations. A more detailed response to your comment on the generalisibility of the model exercise is below.*

An explanation of the choice of modified parameters ("10% of GPP C away from roots and to wood production", "adjusted the turnover time of the SLOW pool, increasing it by 30%") is needed.

*BAE et al: Explanation of these parameter modifications were added.*

*On line 267: "This 10% shift is a conservative estimate of the observed response, where total belowground carbon flux (estimated using a mass balance approach) was ~13% lower in the fertilized watershed (Eastman et al., 2021)."*

*And on line 280: "These reductions in the turnover time of the SOMc/SLOW pool were intended to reflect observed declines in decomposition and increases in POM (Eastman et al., 2022). These declines in decomposition were, in part, a result of a 25-57% in ligninolytic enzyme activity in the fertilized watershed, the primary agent of decomposition of POM (Table 1; Carrara et al., 2018)."*

It also seems critical to me that the model interventions imposed modification of allocation and enzyme inhibition) are contextualised in terms of model applicability (generalisability) to new sites/conditions. In other words, it should be acknowledged that the present paper evaluates only a limited set of predictions and that this does not guarantee that the modified MIMICS-CN model would also perform superior compared to the standard MIMICS-CN and compared to CASA-CN under other experimental manipulations or for other ecosystem types or climatic settings. When imposing the model to capture an observed response and the match between the modelled and observed response is presented as a result, I get the impression of some circularity. The manuscript would benefit from a clarification of the generally valid insights from the modelling exercise and the generalisability of the its predictions.

*BAE et al.: Further discussion on these limitations was added. We try to highlight that with N additions, models often predict increases in NPP, SOM inputs, and soil C without any reduction in decomposition. This is not supported by many experimental studies. We recognize the limitations that you point out here. We also value the modelling exercise of identifying mechanisms controlling N addition responses, and identifying sensitive processes for future model development.*

*On line 375: "We recognize that the same parameter modifications may not lead to the same model-observation improvements at other sites with different ecosystem properties or climate. Nonetheless, this exercise allowed us to identify the potential mechanisms and processes that could be better developed in models to more broadly apply across ecosystems and climates under conditions of elevated N deposition."*

*And lines 463-471:*

*"...our post-hoc adjustment of plant C allocation and microbial decomposition with N enrichment were intended to represent ecosystem responses that are commonly observed in nitrogen enrichment studies (Janssens et al., 2010). This experiment allowed us to identify certain mechanisms and processes in the models that exert strong control over the of the formation and stabilization of SOM and that are influenced by N deposition. However, out results cannot necessarily be generalized to other ecosystems or environmental changes. Future model developments, therefore, should focus on constructing more process-based representations of these mechanisms (i.e. dynamic plant carbon allocation and reduced ligninolytic enzyme activity with N addition) to better predict the often observed reductions in decomposition and soil respiration under N addition that are currently hard to capture with most soil biogeochemical models."*

The manuscript by Eastman et al. represents a relevant attempt to improve current modelling efforts to more adequately reflect responses of forest ecosystems to changes in local nitrogen levels. One of the main outcomes of the study is the significance of including an explicit representation of soil microbial physiology in existing biogeochemical models that link C-N cycles and SOM decay.

My overall impression is that this manuscript presents an interesting study for the biogeochemical community and for future predictions of how changes in nitrogen inputs may alter forest C cycling. The study is well designed and the data is clearly presented. The results support the hypothesis that including realistic estimates of plant biomass partitioning, soil organic matter decomposability and microbial physiology in biogeochemical models can improve predictions of ecosystem responses to environmental change. In particular I enjoyed the discussion and found that the authors did a good job in highlighting the caveats related to the model structures and lack of representation of key soil variables. Yet some aspects of the study still need to be clarified and some terminology should be better explained.

Please note that as an experimentalist, my review is has largely focused on the conceptual aspects of the study more than on the choice and parameterization of the biogeochemical models (namely section 2.2 in the Methods).

*BAE: Thank you for your time spent reviewing this manuscript, and for offering many helpful comments and suggestions to improve the manuscript.*

Main comments:

-The authors talk about plant C allocation throughout their manuscript, although they are mainly referring to shifts in plant biomass partitioning. I suggest they adapt this terminology.

*BAE: We chose to use "C allocation" because the model's allocation of vegetation C includes biomass partitioning and root exudation fluxes. Additionally, shifts in GPP C partitioning can affect plant respiration. We've kept the more inclusive term of C allocation and defined this term in the revised text.*

-When introducing 'microbial physiology' and 'microbial activity', it would be helpful to explicitly provide some examples of the parameters that are commonly used to incorporate these concept into modelling efforts.

*BAE: We have revised the text to the following:*

*"Such microbially explicit models represent microbial physiology through parameterized catabolic processes (e.g., Michaelis-Menten kinetics of decomposition: Vmax, Km) and anaboloic processes (e.g., C use efficiency, N use efficiency, turnover rates)."*

-Line 58: do the authors mean reductions in microbial catabolic activity?

*BAE: Yes. Change made.*

-Line 115: It would be important to specify in which years the observed biomass of trees was measured. At the moment it is quite vague, especially for the fine root assessments. Was this done as in a previous paper by the authors: doi: 10.1111/nph.17256? Moreover, it is not clear if the aboveground and belowground biomass assessments where done for all the dominant plant species present in the forest or only for certain tree species. Since not all plant species respond similarly to nitrogen additions (especially in terms of fine root growth) I think this aspect should be better clarified in the method section and also briefly discussed.

*BAE: Yes, detailed methods are described in a previous paper. However, we have added some more detail to our methods section such as the years in which some key measurements were made and that biomass was estimated for all trees >2.54 cm diameter at breast height. We do not go into details on the differing responses of plant species to nitrogen additions, because this is discussed in depth in several previous studies and is beyond the scope of the model comparison.*

Additional comments:

Some sentences are quite long and could be shortened. E.g. Line 51-54: Consider splitting this long sentence into two. Line 55 – 61: I also found this sentence to be very long, and advise the authors to split it in more than one.

*BAE: changes made to increase clarity and readability.*

Line 57: I assume the authors meant 'fresh inputs of organic matter to the soil'

*BAE: Yes, thank you. Change made.*

Line 60: change 'microbial models' to 'microbially explicit models'.

*BAE: change made*

Line: 239: possible typo: did the authors want to write 'away from roots and towards wood production'?

*BAE: Yes, thank you. Change made.*
* * *
**REVIEWER 2**

GENERAL COMMENTS

Eastman and colleagues used vegetation and soil CN data from a long-term large-scale N fertilization study to evaluate and improve two soil biogeochemical models, coupled to the same (also to be improved) vegetation model. Allowing flexible carbon allocation/partitioning of the vegetation model from fine roots to wood under N enrichment, and increasing POM turnover time in the soil models resulted in simulated N deposition effects on plant C allocation, soil respiration, POM accumulation and soil C:N ratio much closer to the observed values in magnitude and direction.

Something I appreciated about the study is the use of data from such long-term (decades long) N enrichment manipulation, and how the data were not only used to evaluate models but also actually improve them. The manuscript fits well within the scope of Biogeosciences and the special issue.

*BAE: Thank you for your time in reviewing this manuscript, and for the useful comments and suggestions that improved the manuscript.*

I have two major points for improvement, to be addressed before the manuscript can be accepted in my opinion:

- Aspects related to soil layers: is there an organic layer, and if yes, is this simply the litter layer? Are there multiple mineral soil layers and is that relevant to the research questions addressed and the overall aim of the models? Make the existence of layers (if any) clearer at the beginning of M&M where the site

and soil are described, and check the use of the terms "litter layer" vs "organic layer" throughout the manuscript. See also my more detailed specific comments below, including on whether the soil models should be vertically explicit (or not) with different modes of transport in between layers.

*BAE: Thank you for bringing this to our attention. We have gone through the text to clarify the pools being described.*

*For observations, we consider the organic horizon both the litter layer and the organic sub-horizons. We have added text to the methods site description that describes what we are referring to as "organic horizon."*

*In the models, there is one metabolic and one structural litter pool that incorporate both aboveground and belowground plant residues. There are three soil pools (MIC/SOMa, PASS/SOMp, and SLOW/SOMc). There is no layering of surface or soil pools with depth. Rather, they represent five pools of C and N with different turnover times and pathways of decomposition with no spatial equivalency to soil horizons or litter layers.*

*When we calculated total soil C model output, we summed the three soil pools plus the structural and metabolic litter pools. We have combed through the manuscript to adjust our terminology to clearly describe what we are referring to when describing soil or litter outputs from the model.*

*Vertical resolution of soil horizons is a long-term goal of the MIMICS-CN model, but the results presented in this study are just for a single vertical layer (litter + 0-45 cm of mineral soil). While not vertically resolved, both models are validated at the global scale. Considering or incorporating vertical resolution to the models was beyond the scope of this study. However, we do recognize the importance of studying deep soil C storage, and will add to the conclusion a sentence on how including the full soil profile and soil C stock in the future could improve accuracy of the land C sink predictions.*

- Long-term N deposition (at first sight perhaps unexpectedly) increased watershed soil C:N. This can be explained by the increased (high C:N) POM turnover time, and not by increased (high [although reduced] C:N) leaf litter input, in the first place simply because there was no such increased litterfall observed (as opposed to higher N and nutrient addition levels in e.g. doi.org/10.1007/s10021-020-00478-8; doi.org/10.1007/s10342-020-01327-y). I strongly recommend to emphasize this more, in the text but especially by providing data in tables or figures, that clearly show what happened with

observed and simulated litter and soil C, N and C:N, for both the "control" and "N enriched" watersheds. See also my comments on Table 2 and Figure 2.

**BAE: See responses to your suggestions to add additional data/results in the form of a table. In short, if given the opportunity to revise this manuscript, we will add additional data to the appendix with final C and N stocks and C:N ratios.**

SPECIFIC COMMENTS

*Line 30* – "N deposition from the combustion of fossil fuels": this N (mainly NOx), directly emitted through fossil fuel combustion, is in many regions only part of the N deposited. In many regions NH3 from agricultural sources is a greater contributor, although this may be different at your study site.

**BAE: text edited to:**

**"For example, many temperate forests have received decades of reduced N deposition from the combustion of fossil fuels and agricultural sector, which likely released them from N limitation and contributed to significant C sequestration…"**

**And we have added the following citation which shows trends in NOx and NH3 deposition across the US: Li et al., 2016:**
www.pnas.org/cgi/doi/10.1073/pnas.1525736113.

*Line 92* – The site description with experimental manipulation is relatively complete, and it is well explained why it is acceptable to compare long-term N deposition effects between the two watersheds with "only" n = 1. However, what should be explained in more detail and more clearly is the features of the soil, especially in light of the manuscript's focus. I got particularly confused on whether there is an actual organic layer. According to Lines 118-119 there were "organic horizon C and N stocks" measured, but from the broader context of the manuscript and model description I interpret this as a litter layer and that there would be no F or H layers, nor Ah? --> please mention in the first paragraph of the methodology if and how the organic and mineral soil are layered in the upper 45 cm, including layer depths for both watersheds if not too much within-watershed variation.

**BAE: This description has been added to the materials and methods. See also our response to your first major comment above.**

*Lines 118-119* – The "organic horizon C and N stocks": interpreting these as the litter pools (but also if it's an H, .. layer), I think these should be mentioned not only for Control observations and simulations (e.g. "organic horizon C" in Table 2), but also for N deposition observations and simulations. See also my further comments on display items.

**BAE: The terminology has been adjusted and a more detailed description of what horizons are included in the litter + top 45 cm of soil has been added. See above responses.**

*Lines 171-172* – "stoichiometries of litter … are a prognostic feature of the model, influenced by various factors including litter stoichiometry" Seems like this expression is circular. Please check.

**BAE: Text changed to:**

**"The stoichiometries of SOM pools, however, are a prognostic feature of the model that reflect litter chemistry, microbial necromass inputs, and the relative abundance of different SOM pools."**

*Line 239* – There was an allocation/partitioning shift from roots to wood production, but no shifts in leaf litter inputs to soil, I understand. Was there also no increase in LAI under N deposition?

**BAE: We do not have reliable data on LAI from the study site. And, yes, it is true that we found no differences in leaf litter inputs to the soil between the fertilized and reference watersheds.**

*Line 274* – Again, unclear use of the terms "organic horizon" and "litter layers", and whether these are part of what you consider as "soil" or not. According to the model description (Fig. 1), they are not, but from Table 2 I interpret that "total soil" refers to this "litter layer"/"organic horizon" + "mineral soil layer(s)".

**BAE: See above responses. Terminology adjusted to be more descriptive here and throughout.**

*Line 352* – Good idea to discuss limitations of the models used, and mention some alternatives.

**BAE: Thank you.**

*Table 2* – I suggest to provide all these (final) observed and simulated values for both the reference and manipulated watershed, and for all three model versions ("experiments"). It is further not entirely clear to me why C stocks are given for the organic and mineral horizons as well as their sum, but for instance C:N ratio was only given for the mineral soil here and not the organic soil and their sum. I am aware that making this table with both treatments, all three models + observations and more response variables will make the table much larger, so you may consider providing the table that I suggest here in the supplement/appendix. An alternative could be to provide these observed and estimated pools and fluxes in one or multiple diagrams such as figure 1.

**BAE: Table 2 shows baseline model values, which are not the final modelled values with and without +N. Given the opportunity to revise the manuscript, we will consider adding a table in the appendix with final values from multiple models and observations. Like you said, two watersheds plus two models * two treatments * three versions would be a lot to show in one table. We will consider which would be most valuable to share with the readers, such as final model results from experiment 3 (allocation shift and enzyme inhibition).**

*Figure 1* – Relating to my other comments: both soil biogeochemical models distinguish multiple litter and soil pools, but further the models are not vertically spatially explicit. Does that mean that they can not be used for soils with an organic layer between the litter and mineral soil, or clearly distinguished mineral soil layers? Or is this irrelevant to the research questions that are to be addressed with such models? Or can they easily be adapted with layers with "diffusion" and other mixing transport in between, if needed? I am aware that the models should be "as simple as possible, but as complex as necessary" with respect to the questions they should address, so I'm also just wondering out of curiosity.

**BAE: The different SOM pools defined by the model and density fractionation experiments are just a couple of examples of how to separate SOM into functionally different classes with different stoichiometries and turnover times. Neither model distinguishes an organic layer between the litter and mineral soil. To incorporate this would require additional microbial, chemically protected and physically protected pools for this layer which may have different soil temperature and moisture functions. Adding those organic layer pools would require substantial code and input file modifications. In some forests, modeling the organic layer may be important for computing C stocks, but was not considered for this study.**

**Neither of the models are vertically resolved but consider one vertical soil layer (0-45 cm in this case). Some efforts have been made in the modelling community to**

*parameterize vertical resolution of soil, such as in the Community Land Model (CLM), which could potential help model agreement with observed soil C stocks, and changes in soil C to global change (e.g., Koven et al., 2013 **doi:10.5194/bg-10-7109-2013**; Riley et a., 2014 doi:10.5194/gmd-7-1335-2014). Nonetheless, even the vertically resolved CLM model underestimates turnover times of deeper soils and likely needs to incorporate other SOM stabilization properties such as microbial activity, mineral associations, and root-soil feedbacks to improve observational agreement (Koven et al., 2013).*

*Figure 2* – The figure presents pool and flux responses to N enrichment. For the soil responses, it is clear what are the pool and flux responses, but for the vegetation responses it is not entirely clear: ANPP is a flux, but does "leaf C" represent litterfall, "wood C" wood production, and "fine root C" fine root production (so all fluxes)? I suggest to make this clear in the figure itself, or at least a little more clear than it is now in the caption.

**BAE: Thank you. Leaf, wood, and root C are pools. We will change the figure axis labels and legend to make whether each component is a pool or flux clearer.**

*Figure 2* – "total soil C and C:N" = litter layer and mineral soil layer combined?

 **BAE: That is correct. We will add this description to the legend.**

TECHNICAL CORRECTIONS

*Line 38* – "and forest soils recover"

**BAE: change made**

*Line 93* – January temperature should be -1.8 °C, I think.

**BAE: That is correct. Change made.**

*Line 258* – response ratio as ambient/+N. Shouldn't this be +N/ambient?

**BAE: That is correct. Change made.**

*Line 416* – "forest ecosystems"

**BAE: change made**

*Line 435* – "composition"

*BAE: change made*

*Line 435* – "modeling" with one "l" in American English, as used elsewhere in the manuscript. Check also use of "modelling" elsewhere in the text.

*BAE: change made here and throughout*

*Line 701* – "increasing the turnover time"

*BAE: "decreasing" is correct here. For CASA, the first-order turnover time was increased to reduce decomposition rates. In MIMICS-CN this same change was achieved by increasing the half-saturation constant.*

*Line 706* – "modeled responses"

*BAE: "observed and modelled response ratios" was our intended text here. No changes made.*

**REVIEWER 3**

The authors conducted an interesting model-experiment integration study to explore the soil carbon (C) response to long-term nitrogen (N) enrichment. I appreciate that the authors attempt to improve the representation of plant-soil interactions and their responses to N additions, which is still challenging, despite the development and use of C-N coupled models for many years.

*BAE: Thank you for your time in reviewing this manuscript, and for the useful comments and suggestions that improved the manuscript.*

The manuscript can be further improved by addressing the following concerns:

1. While this study focuses on the ecosystem responses to N addition by presenting the response ratio, it would be beneficial to highlight the advantage of using the 30-year long-term experiment. For example, are there any long-term temporal data available for model testing or validation? Furthermore, the modeling results should also include uncertainty quantification (e.g., in Fig. 2) if possible.

*BAE: We appreciate and agree that long-term data provide unique opportunities to study environmental change and inform models. We have added the following sentence to the introduction where the long-term study is introduced: "Long-term experimental manipulations at relatively large scales (e.g., watersheds or large forested plots) are rare but important because significant ecosystem processes can respond slowly to sustained changes in their environment" (e.g., changes in species composition of herb-layer and forest trees; soil chemistry due to buffering reactions).*

*If we understand correctly, you are asking for comparing temporal (time series?) data to model responses? While we do have some repeated measurements of leaf litter and woody biomass, these don't include C and N concentrations. Other measurements were made at varying scales and using varied methods, which make it hard to string together into a time series. This paper focuses on the directional responses of ecosystem variables to +N, using the models to test hypotheses about the controlling mechanisms, and evaluating the adequacies or inadequacies of the relevant model processes. For this study, we consider the 30-year response of the forest or simulated forest a good benchmark for evaluating these objectives.*

*If we understand your final comment correctly, including uncertainty quantification of the models would require additional parameter sensitivity tests to generate uncertainty estimates around the modeled responses. Towards this end, we hope to make it easier to calibrate ecosystem models with observational data (e.g., Pierson et al.,2022: https://doi.org/10.1038/s41598-022-14224-8). And while we recognize this is a good idea, additional analyses trying to quantify model parametric uncertainty are outside the scope of this manuscript.*

2. The authors should be cautious regarding their claim on "MIMICS-CN offers an advantage over CASA-CN". The actual "advantage" is not clearly demonstrated, given that MIMICS-CN only improves the total soil C:N compared to CASA-CN, not to mention the large uncertainty in the observed response of total soil C:N. In addition, a common issue faced in microbial modeling is the lack of microbial data, which greatly weakens the value and results of the microbial model used herein. Without microbial data, the microbial model remains a black box as no microbial mechanisms can be explicitly revealed to advance our understanding the key processes.

*BAE: Thank you for this comment. We have changed the statement to "offers a potential advantage..." We also highlighted how at our sites and others, an apparent shift in microbial composition (i.e., less fungal decomposers) contributes to reduced decomposition of high C:N SOM:*

*"Additionally, the microbial community composition, as approximated by the relative abundance of MICr:MICK simulate by MIMICS-CN, was not sensitive to N additions or shifts in plant allocation and inputs (not shown). By contrast, N addition experiments in forests ecosystems have found reductions in fungal decomposer biomass, reduced ligninolytic enzyme activity, (Frey et al., 2014; Argiroff et al., 2019), and a shift in community function with reduced ability to decompose recalcitrant SOM (Ramirez et al., 2012)—including at our study site (Carrara et al., 2018; Moore et al., 2021)."*

*Furthermore, our discussion section goes into depth about the limitations and need for constraining microbial processes with observations.*

3. I acknowledge that the authors state this model experiment is a post-hoc modification to both models. I would agree such a post-hoc modification is valuable. However, even with the knowledge informed by the observations, both models still failed to improve the modeling of the relationship between bulk soil C:N ratio versus the fraction of soil C in POM (Fig. 3). A more in-depth discussion and practical suggestions should be provided on this topic.

*BAE: Although we already offer several suggestions to address this shortcoming throughout the discussion, these could include:*

- *Better parameterization of parameters for which data do already exist. See line 425: "Specifically, constraining carbon use efficiencies (CUE), nitrogen use efficiencies (NUE), and C:N ratios for microbial communities against data and observations is warranted to capture their responses to environmental changes."*
- *More explicit representation of rhizosphere priming that targets POM and is downregulated with N additions. See line 352: "One shortcoming of our model efforts was the inability to represent a meaningful rhizosphere priming response in the control simulation that would lead to reduced priming in the elevated N simulations."*
- *And the need for more feedbacks between the N and C cycles (up- or down-regulation of some processes based on N status; lines 360, 407).*

- I also recommend to consider the the large uncertainty in soil C:N observations when drawing conclusions about general model comparison of the "microbial-explicit" (MIMICS-CN) vs. the "microbial-implicit" (CASA-CN) model. In this sense, the statement in the abstract ("With all of these modifications, only the microbially explicit model

captured a positive soil C stock and C:N response in line with observations.") could be misunderstood as soil C:N and soil C stocks offering a constraint for model selection. However, all model variants lie within the range of uncertainty of observations (Fig. 3). Such a misunderstanding should be avoided. As shown here, the value of the MIMICS-CN model lies, in my reading, in particular in its ability to predict a soil C:N increase under N fertilisation (when additionally considering the imposed enzyme inhibition).

***BAE et al.: The language in the abstract and throughout the methods and discussion has been changed to accurately describe when model predictions fell within or outside of the observed range (mean +/- 1 se). E.g., see revised manuscript around lines 322 and 349.***

- Furthermore, the hypothesis stated around line 82 should be explained and motivated in the introduction.

***BAE et al: This first hypothesis was reworded to be better explained (text underlined is new): "We hypothesized that the default models (experiment 1)*** ***would be N limited and, thus,*** ***respond to N additions with a reduction in N limitation, increase in plant productivity, and subsequent increase in soil respiration and soil C*** ***stocks as plant C production and inputs increase******."***

***Additionally, the third paragraph of the introduction has been revised to more directly motivate this hypothesis by comparing models and theory of microbial responses to N availability and the inclusion or absence of plant-microbial interactions in microbially explicit models (see revised manuscript lines 59-70).***

- Text around line line 353 would benefit from an explanation of the link between exudation, mycorrhizal colonisation, and the allocation or the enzyme inhibition "tweak" of the model.

***BAE et al: We added the following text to this paragraph:***

***"When we shifted in the overall C allocation of plants (through a parameter change in the second "allocation shift +N" experiment) to increase wood production and reduce root production, this parameter change did reduce soil respiration relative to the control run, but does not account for all mechanisms that reduce soil respiration. Rather than an overall reduction in decomposition and accumulation of POM (as observed), this allocation shift reduced litter inputs from roots to the soil, and thus a***

*relative decrease in total soil C (Fig. 2)."*

I am therefore returning the paper to you so that you can make the necessary (major) revisions and I am looking forward to receiving your revised manuscript.

Beni Stocker

---

## Author Response (AR2)

**BAE et al: Dear Referees, Thank you kindly for reviewing the manuscript a second time and catching a few additional changes that could be made to better this manuscript. Our specific responses are below in bold text.**

Anonymous Referee #3 suggestions for revision (August 24, 2023)

I would like the authors to reconsider my previous suggestion on the quantification of modeling uncertainty because (1) the one-value modeling result is vulnerable, and (2) it's not difficult to quantify uncertainty (e.g., generating error bars) for the modeled results in Fig. 2.

**BAE et al: Error bars were added to Fig. 2 from propagating the standard error representing the interannual variability of modeled pools and fluxes in the last ten years of the experimental simulations.**

Anonymous Referee #2 suggestions for revision (September 4, 2023)

GENERAL COMMENTS

The authors addressed the comments raised by myself and other reviewers sufficiently, in my opinion (e.g. on modified parameterizations vs model structure, (organic) soil layers in situ vs models, generalizability). I therefore recommend acceptance of this manuscript, pending a few technical corrections to be made.

SPECIFIC COMMENTS

Line 746 – Should be "increasing the turnover time (…)", as I suggested in the previous round of review? : increasing/longer turnover time implies slower turnover, e.g. Table 1.

**BAE et al: Thank you, change made.**

Table A4 – Second line for vegetation is referred to as NPP here, but represents ANPP with an equal observed value in Table 2. Please check and correct.

**BAE et al: The modeled values are NPP, and observed ANPP, in this table. This is noted in the footnote of the table (A4).**

TECHNICAL CORRECTIONS

Line 268 – "increases in the turnover time" ?

**BAE et al: Change made.**

Line 270 – "a 25-57% reduction in ligninolytic enzyme activity"

**BAE et al: Change made.**

Line 790 – "modeled" (one "l") in American English? Also edit elsewhere in the manuscript if applicable.

**BAE et al: Change made.**

---

## Author Response (AR3)

**Author Responses to Associate Editor Review**

Dear Dr. Eastman

Thank you for your patience and for addressing the second set of review comments. I have noticed four remaining technical points which I kindly ask you to resolve before I can accept your manuscript for publication. These are:

- Please add an explanation of the error bars, which you've added in your latest revisions, in the caption of Fig. 2.

**BE et al: The explanation of the error bars is included in the full figure caption on lines 755-766**

- Fig. 2, there is a cross symbol next to the 'Soil Respiration' label. Could you please add an explanation of its meaning or remove it?

**BE et al: The explanation of the symbol is included in the full figure caption on lines 763-766**

- The text in the 'Conclusions' section should be revised in two places. On l. 473, should 'led to' be added to complete the sentence? ("notably, a shift in plant C allocation [...], decreased soil respiration, and *led to* an accumulation of POM"). On l. 474, the following sentence doesn't seem to make sense: "Our results also suggest that while a microbially explicit model has potential to incorporate additional plant-microbe processes or better parameter existing processes, [...]". Please revise.

**BE et al: Change made on line 473. The following sentences were revised to the following:**

**"Our results also suggest that a microbially explicit model has greater potential than a microbially implicit model to incorporate additional plant-microbe processes and better parameterize existing processes because of existing plant-microbial processes include in the model structure. However, testing model experiments at additional sites and additional data is required to improve model representation of the complex plant-microbial responses to N enrichment and model predictions of ecosystem recovery following N additions."**

- Table A1: Avoid abbreviations or explain abbreviations and model variable names in the caption (e.g., 'xkNlimiting'). Provide units of C:N ratios (mass or molar units?)

**BE et al: Changes made.**

Please revise these points and resubmit a corrected version of your manuscript.

Thank you for for continued efforts and your contribution to our Special Issue.

**BE et al: Thank you.**

Best wishes,

Beni Stocker